# Anti-Inflammatory Oxysterol, Oxy210, Inhibits Atherosclerosis in Hyperlipidemic Mice and Inflammatory Responses of Vascular Cells

**DOI:** 10.3390/cells13191632

**Published:** 2024-09-30

**Authors:** Frank Stappenbeck, Feng Wang, Satyesh K. Sinha, Simon T. Hui, Lia Farahi, Nigora Mukhamedova, Andrew Fleetwood, Andrew J. Murphy, Dmitri Sviridov, Aldons J. Lusis, Farhad Parhami

**Affiliations:** 1MAX BioPharma Inc., Santa Monica, CA 90404, USA; fstappenbeck@maxbiopharma.com (F.S.); fwang@maxbiopharma.com (F.W.); 2Department of Medicine, Division of Cardiology, David Geffen School of Medicine, University of California, Los Angeles, CA 90095, USA; sksinha@mednet.ucla.edu (S.K.S.); sthui@mednet.ucla.edu (S.T.H.); lfarahi@g.ucla.edu (L.F.); jlusis@mednet.ucla.edu (A.J.L.); 3Baker Heart and Diabetes Institute, Melbourne, VIC 3004, Australia; andrew.fleetwood@baker.edu.au (A.F.); andrew.murphy@baker.edu.au (A.J.M.); dmitri.sviridov@baker.edu.au (D.S.); 4Department of Biochemistry and Molecular Biology, Monash University, Clayton, VIC 3168, Australia

**Keywords:** atherosclerosis, NAFLD/NASH, MAFLD/MASH, oxysterols, Oxy210, APOE*3-Leiden.CETP mouse model, chronic inflammation, fibrosis, endothelial cells, macrophages

## Abstract

Background and aims: We previously reported that Oxy210, an oxysterol-based drug candidate, exhibits antifibrotic and anti-inflammatory properties. We also showed that, in mice, it ameliorates hepatic hallmarks of non-alcoholic steatohepatitis (NASH), including inflammation and fibrosis, and reduces adipose tissue inflammation. Here, we aim to investigate the effects of Oxy210 on atherosclerosis, an inflammatory disease of the large arteries that is linked to NASH in epidemiologic studies, shares many of the same risk factors, and is the major cause of mortality in people with NASH. Methods: Oxy210 was studied in vivo in APOE*3-Leiden.CETP mice, a humanized mouse model for both NASH and atherosclerosis, in which symptoms are induced by consumption of a high fat, high cholesterol “Western” diet (WD). Oxy210 was also studied in vitro using two cell types that are important in atherogenesis: human aortic endothelial cells (HAECs) and macrophages treated with atherogenic and inflammatory agents. Results: Oxy210 reduced atherosclerotic lesion formation by more than 50% in hyperlipidemic mice fed the WD for 16 weeks. This was accompanied by reduced plasma cholesterol levels and reduced macrophages in lesions. In HAECs and macrophages, Oxy210 reduced the expression of key inflammatory markers associated with atherosclerosis, including interleukin-1 beta (*IL-1β*), interleukin-6 (*IL-6*), tumor necrosis factor-α (*TNF-α*), chemokine (C-C motif) ligand 2 (*CCL2*), vascular cell adhesion molecule-1 (*VCAM-1*), and *E-Selectin*. In addition, cholesterol efflux was significantly enhanced in macrophages treated with Oxy210. Conclusions: These findings suggest that Oxy210 could be a drug candidate for targeting both NASH and atherosclerosis, as well as chronic inflammation associated with the manifestations of metabolic syndrome.

## 1. Introduction

Atherosclerosis and its complications are the underlying cause for most heart attacks and strokes, responsible for significant morbidity and mortality worldwide [1]. A chronic inflammatory disease of the vascular wall, atherosclerosis involves endothelial cell dysfunction, accumulation of lipids, namely low-density lipoprotein cholesterol (LDL-C), in the subendothelial space, modifications of LDL-C (e.g., aggregation, oxidation and glycation), increased adherence of monocytes to endothelial cells and their migration into the subendothelial space, formation of foam cells by macrophages and smooth muscle cells that engulf the accumulating lipids, and the progression of the early fatty streak to a lipid filled atheroma containing a necrotic core and cholesterol crystals [1,2,3]. As atherosclerosis combines features of lipid storage as well as chronic inflammatory disorders, therapeutic strategies have been devised accordingly over the years; any effective strategy should ideally address both aspects [3,4,5].

Among drugs currently in use to treat atherosclerosis, statins, such as Atorvastatin (Lipitor), reportedly exhibit clinically significant anti-inflammatory effects in addition to their lipid-lowering properties [6]. However, the anti-inflammatory effects appear as a downstream consequence of lipid lowering and therefore lack synergy [7], even though some reports of direct anti-inflammatory effects of statins in vitro have been published [8]. Unfortunately, the use of statins and other lipid-lowering drugs could only reduce but not eliminate the incidence of atherosclerosis, and these drugs are plagued by several complications that limit their use and patient compliance [9]. In 2017, the results of the CANTOS trials were published [10], documenting significant clinical benefits of Canakinumab, an interleukin-1 beta (IL-1β) neutralizing antibody, in the context of cardiovascular disease (CVD), which sparked renewed enthusiasm for the examination of anti-inflammatory therapies, including Colchicine and Methotrexate, in atherosclerosis [1,2,3,4,5,11]. However, none of these established drugs can combine powerful anti-inflammatory properties with meaningful lipid-lowering activity in a synergistic manner.

Classic risk factors for atherosclerosis include diabetes, hypercholesterolemia, hypertension, and smoking. In addition, some bacterial infections, including gingivitis [12], and exposure of the vascular endothelium to advanced glycation end products may increase cardiovascular risk [13]. Hence, the modification of these risk factors, through pharmaceutical and non-pharmaceutical means, have shaped efforts to combat atherosclerosis and CVD over many years. Despite all improvements in disease management, mortality from atherosclerosis is expected to increase across the world due to a “changing landscape” of the disease [1], tied to demographic changes and increasing rates of obesity observed globally.

In addition to the classic risk factors, non-alcoholic fatty liver disease (NAFLD)/NASH, also known as metabolic-associated fatty liver disease (MAFLD)/metabolic dysfunction-associated steatohepatitis (MASH), recognized as the liver pandemic of our century [14,15], can contribute to atherogenesis in multiple ways, for example, by increasing oxidative stress and the release of pro-atherogenic molecules in the vasculature, such as tumor necrosis factor-α (TNF-α), interleukin-6 (IL-6), IL-1β, and oxidized LDL-C, resulting in atherogenic dyslipidemia, macrophage activation, and chronic injuries to the vascular endothelium [16,17,18,19,20]. Human epidemiologic studies and studies in animal models have demonstrated significant correlations between NAFLD/NASH and atherosclerosis, with NAFLD being an apparent risk factor for atherosclerosis independent of other metabolic risk factors [17,18]. Patients suffering from NASH are at increased cardiovascular risk and die from CVD more frequently than the general population or from liver disease. Hence, therapies designed to prevent or treat NASH may also have a preventative and/or therapeutic impact on atherosclerosis [21,22].

Atherosclerosis and NASH share aspects of pathological lipid deposition and macrophage-driven chronic inflammation in a partially overlapping manner. Therefore, new drug treatments for NASH should ideally combine lipid-lowering with anti-inflammatory features in circulation and in the liver while displaying antifibrotic properties in the liver, a challenging proposition. Among approved and investigative NASH drugs [23], some, but not all, fall into these categories. As the first FDA-approved treatment, Resmetirom, a selective thyroid hormone receptor β agonist, reduced circulating lipids and lipoproteins (LDL-cholesterol, apolipoprotein B, TGs, and lipoprotein (a)) and resolved inflammation and hepatocyte ballooning in the livers of NASH patients during clinical trials [24]. It is not known whether Resmetirom therapy has disease-modifying effects in atherosclerosis, although statistically significant reductions in atherogenic lipids were reported in human clinical trials [25].

By contrast, activation of the Farnesoid X receptor (FXR) in the liver, using various small molecule FXR agonists, has been mechanistically tied to development of an undesirable and potentially pro-atherogenic circulating lipid profile [26], despite beneficial FXR-mediated effects detected in livers of NASH patients during clinical trials [27].

Oxysterols are oxidized derivatives of cholesterol with diverse biological properties that differ from those of parent cholesterol. While a causal role for excess cholesterol in atherosclerosis [28] and NASH [29] is well documented, and endogenous pro-inflammatory oxysterols have been implicated in atherogenesis [30], the contributions of endogenous oxysterols in liver disease and, specifically, the transition of simple steatosis to NASH, have, until recently, been less thoroughly studied [31,32,33]. Historically, endogenous oxysterols were often lumped into one group and collectively regarded either as passive and transient metabolites or, more often in the context of atherosclerosis, viewed as all pro-oxidant and pro-inflammatory [34,35]. Importantly, however, oxysterols are “not all the same”, but their individual properties do vary significantly based on chemical composition, and small structural changes can have an outsized impact on properties in this chemical class. For example, small changes to the molecular structure can promote shifts from agonist to antagonist properties of oxysterols active in several cellular signaling pathways, such as inflammatory signaling, including TLR signaling [36], Hedgehog (Hh) signaling, as reported by us [37,38] and others [39], as well as N-methyl-D-aspartate receptor signaling [40]. According to published reports, 7-ketocholesterol, 7α-hydroxycholesterol, and 27-hydroxycholesterol display significant pro-inflammatory and pro-atherogenic properties, known to exacerbate atherosclerosis [41] and NASH [42]. Similarly, 25-hydroxycholesterol has been shown to enhance inflammatory signaling in hepatocytes [43,44], whereas its sulfate derivative, 25-hydroxycholesterol-3-sulfate, displays significant anti-inflammatory properties [44,45], consistent with an active role of these endogenous oxysterols in complex cellular signaling. In fact, 25-hydroxycholesterol-3-sulfate (also known as Larsucosterol or DUR-928) has entered clinical development as a potential NASH therapeutic [46,47]. Although potentially promising due to its lipid-lowering properties [47], it is not known at this time whether Larsucosterol treatment can ameliorate atherosclerosis in mice or humans.

In this report, we describe anti-atherogenic effects of Oxy210, a non-endogenous (man-made) oxysterol-based drug candidate, which we previously evaluated as a prospective NASH therapeutic using APOE*3-Leiden.CETP mice, a mouse model for NASH and atherosclerosis in which symptoms are induced by consumption of a high-fat high-cholesterol “Western” diet (WD) [48,49]. The APOE*3-Leiden.CETP model is characterized by substantial genetic overlap with human NASH and atherosclerosis [49,50,51]. By combining the human cholesteryl ester transfer protein (CETP) transgene and the APOE*3-Leiden transgene, which decreases the clearance of triglyceride (TG)-rich lipoproteins, APOE*3-Leiden.CETP mice exhibit a human-like lipoprotein profile and replicate human atheroma with all five stages of disease represented (type I to V) [52,53]. In response to drug therapies with statins, fibrates, ezetimibe, and antidiabetic medications, these mice demonstrate improved lipid profiles and atherosclerotic progression that resemble those observed in humans [50,51,52,53]. APOE*3-Leiden.CETP mice were also employed to demonstrate the efficacy of anti-PCSK9 therapies for lowering LDL-C and TGs, which translated to reduced atherosclerosis development in the aortic root [53]. In addition, APOE*3-Leiden.CETP mice are recognized as a suitable model of WD-induced metabolic syndrome and its downstream consequences [50,51,52,53].

Oxy210 given in the WD (4 mg/g) was shown to alleviate symptoms of NASH and white adipose tissue inflammation, associated with lowered circulating inflammatory cytokines and significant reductions in the cholesterol burden in mouse plasma, both with total and unesterified cholesterol [36,48]. Here, we demonstrate that oral administration of Oxy210 formulated in the WD over a 16-week period inhibits atherosclerotic lesion formation by more than 50%, associated with a significant reduction in plaque macrophage proliferation (compared to WD control). Furthermore, and in follow up with previous findings [36,48], we also report direct anti-inflammatory effects of Oxy210 when applied in vitro to primary human aortic endothelial cells (HAECs) and macrophages stimulated with atherogenic oxidized phospholipids, inflammatory cytokines, Toll-like receptor (TLR) agonists, or lipopolysaccharide (LPS). Given its disease-modifying properties observed in vivo and in vitro, Oxy210 appears to be a promising candidate for therapeutic development to target both NASH and atherosclerosis.

## 2. Materials and Methods

### 2.1. Cell Culture and Reagents

RAW264.7 mouse macrophages and THP-1 human monocytes were purchased from American Tissue Type Culture Collection (ATCC, Rockville, MD, USA) and cultured in DMEM or RPMI containing 10% fetal bovine serum (FBS) and antibiotics, respectively, as previously reported [36]. THP-1 culture medium also contained 0.05 mM β-mercaptoethanol. THP-1 differentiation into the macrophages was achieved as described [36]. HAECs were purchased from Thermo Fisher Scientific (Waltham, MA, USA) and cultured in M199 containing 10% fetal bovine serum (FBS), large vessel endothelial supplement (LVES, 1X, Gibco), and antibiotics. For experiments, the LVES was removed, and the FBS concentration reduced to 0.1% or 1% at the time treatments were initiated. For cell-counting assays, RAW264.7 cells cultured in DMEM containing 1% FBS in 12-well plates at 20% confluence were treated with Oxy210 for 48 h and then detached from wells by scraping, spun down, resuspended in fresh medium, and counted using a hemocytometer under a light microscope.

LPS and Phorbol 12-myristate 13-acetate were purchased from Sigma-Aldrich (St. Louis, MO, USA), recombinant human-transforming growth factor beta 1 (TGF-β1) from R&D Systems (Minneapolis, MN, USA), MPLAs from InvivoGen (San Diego, CA, USA), 1-palmitoyl-2-glutaroyl-sn-glycero-3-phosphorylcholine (PGPC) and CU-T12-9 from Cayman Chemical (Ann Arbor, MI, USA). Oxy210 was prepared by MAX BioPharma, according to a previously reported procedure [54].

### 2.2. Quantitative RT-PCR

Total RNA was extracted with the RNeasy Plus Mini Kit from Qiagen (Hilden, Germany), according to the manufacturer’s instructions. One microgram of RNA was reverse-transcribed using an iScript Reverse Transcription Supermix from Bio-Rad Laboratories (Hercules, CA, USA) to make single-stranded cDNA. The cDNAs were then mixed with Qi SYBR Green Supermix (Bio-Rad) for quantitative RT-PCR assay using a Bio-Rad I-cycler IQ quantitative thermocycler. All PCR samples were prepared in triplicate wells in a 96-well plate. After 40 cycles of PCR, melt curves were examined in order to ensure primer specificity. Fold changes in gene expression were calculated using the ΔΔCt method. Primer sequences used for mouse and human genes are shown in Appendix A.

### 2.3. VCAM-1 Assay

The human VCAM-1 ELISA kit was purchased from Thermo Fisher Scientific (Waltham, MA, USA). ELISA was performed according to the manufacturer’s instructions. Briefly, cells were cultured in 12-well plates and treated with the test reagents for 72 h and then lysed in 100 μL of cell lysis buffer. Cell extracts were suitably diluted and used in the assay together with the serially diluted Human VCAM-1 protein standard solution. The protein concentration was measured using Bio-Rad Protein Assay reagent (Bio-Rad, Hercules, CA, USA).

### 2.4. Cholesterol Efflux

RAW264.7 cells were grown to confluence in a complete medium with 10% fetal calf serum (FCS). Cholesterol efflux was assessed as described previously [55]. In brief, cellular cholesterol was labeled by incubating cells in a medium containing 10% FCS and [1a,2a(n)-^3^H] cholesterol (American Radiolabel Chemical Incorporation, St. Louis, MO, final radioactivity 0.5 MBq/mL) for 48 h in a CO_2_ incubator. Cells were washed and incubated for 24 h with Oxy210 (5 µM), or vehicle, in the medium with 0.1% FCS. Next cells were washed and incubated for 2 h at 37 °C in serum-free medium containing 25 μg/mL of lipid-free human apoA-I (CSL). The medium was collected, centrifuged for 15 min at 4 °C at 2000× *g*, and aliquots of supernatant were counted in a β-counter. Cells were harvested, and cell-associated radioactivity was counted. Specific cholesterol efflux was expressed as the proportion of [^3^H] cholesterol transferred from cells to medium minus that in the absence of apoA-I.

### 2.5. Western Blotting

RAW264.7 cells were incubated with Oxy210 (5 µM), or vehicle, for 24 h in medium with 0.1% FCS. Cells were then lysed, proteins separated on 4–12% SDS-PAGE and ABCA1 was detected by anti-ABCA1 antibody (H10, Abcam, Cambridge, UK) and visualized and quantitated using G: Box imaging system.

### 2.6. Animal Studies

The breeding and characterization of transgenic mice expressing human CETP and the human APOE*3-Leiden (E3L) were described previously [49]. To generate mice for the atherosclerosis studies, male C57BL/6J mice carrying both transgenes were bred with BXD19/TyJ females, and F1 progeny carrying both transgenes were used. Animals were maintained on a 12 h light–dark cycle with ad libitum access to food and water. All diets were formulated by Research Diets, Inc. (New Brunswick, NJ, USA). Control mice were fed a WD (33 kcal% fat from cocoa butter and 1% cholesterol, Research Diets, cat# D10042101) for 16 weeks, whereas Oxy210-treated mice were fed a WD supplemented with Oxy210 at 0.5, 1, 2, or 4 mg/g of food. Food consumption was monitored, and there were no significant differences in body weights at the end of the study, except for a small but significant reduction in weight gain in the 4 mg/g cohort [48]. No evidence of toxicity during the 16-week treatment with Oxy210 was noted as mice exhibited normal behaviors, grooming, eating, and physical activities. Necropsies showed no evidence of Oxy210 toxicity to any organs. All animal work was approved by the UCLA Animal Research Committee, the IACUC.

### 2.7. Plasma Lipids

Mice were fasted for 4 h. Plasma lipids were measured by colorimetric analysis as previously described [48,49].

### 2.8. Quantification of Atherosclerotic Lesions and Macrophage Proliferation

As previously reported [56,57], following euthanization, the animal’s chest cavity was opened, and PBS was used to perfuse the vasculature. The upper portion of the heart and proximal aorta were then collected, embedded in OCT compound, and stored at a temperature of −70 °C. Cryosections of the aorta, measuring 10 μm in thickness, were taken in a series, starting from the aortic root, and placed on poly-D-lysine-coated slides. These sections were subsequently stained with oil red O and hematoxylin. To capture the images, a Nikon Eclips microscope was utilized. From each mouse, ten sections were counted at intervals of 120 μm. The lesion area was quantified using ImagePro Premier software (Version 9.1) from Media Cybernetics, Inc. (Rockville, MD, USA). This study strictly followed the guidelines outlined in the American Heart Association Statement for experimental atherosclerosis studies. To quantify macrophage content and proliferation in lesions, immunofluorescence staining was conducted following a standard protocol [57,58]. Briefly, frozen aortic sections measuring 10 μm in thickness were brought to room temperature and fixed with 4% paraformaldehyde. Subsequently, the sections were permeabilized using 0.1% Triton X-100 and blocked with 5% normal goat serum and 3% bovine serum albumin. Next, the sections were incubated overnight at 4 °C with anti-Ki67 antibody (dilution 1:750, abcam, Cambridge, MA, USA) to assess proliferation [57,58,59], in conjunction with anti-CD-68 antibody (dilution 1:400, BD Biosciences, San Jose, CA, USA) to identify macrophages. After washing with phosphate-buffered saline (PBS), corresponding secondary antibodies (goat anti-rabbit (Alexa Fluor^®^ 488, Life Technologies, Waltham, MA, USA) and goat anti-rat (Alexa Fluor^®^ 594, Life Technologies)) were applied for 1 h at room temperature to detect antibody to CD68 or Ki67. Coverslips were mounted using fluoroshield containing DAPI (Sigma, St. Louis, MO, USA). The quantification of Ki67+ staining was performed by normalizing the Ki67+ nuclei colocalized with intracellular and membrane-associated CD68+ components to total nuclei numbers in the CD68+-stained area in lesions using a Nikon (Eclipse Ti-s) microscope (Melville, NY, USA)

### 2.9. Statistical Analysis

Statistical analyses were performed using the StatView 5 program (SAS Institute, Cary, NC, USA). All *p*-values were calculated using ANOVA and Fisher’s projected least significant difference (PLSD) significance test. A value of *p* < 0.05 was considered significant. The IC_50_ dose–response curves were modeled using a five-parameter logistic model. This model allows for asymmetric curves and automatically estimates the mean maximum and minimum response. Based on this model, the IC_50_ values were estimated, corresponding to the dose halfway between the minimum and maximum response. Models of dose versus response and dose versus log response were also evaluated. The R square statistic was computed as a measure of model fit.

## 3. Results

### 3.1. Oxy210 Reduces Atherosclerosis and Macrophage Proliferation and Content in Atherosclerotic Lesions in APOE*3-Leiden.CETP Mice

As previously reported, Oxy210 induced a combination of anti-inflammatory effects (in hepatic and adipose tissue) and lipid-lowering effects (total and unesterified plasma cholesterol) in APOE*3-Leiden.CETP mice on the WD [36,48]. We next examined the potential anti-atherogenic effects of Oxy210 in this same mouse model. Oral administration of Oxy210 formulated in WD at 4 mg/g for 16 weeks showed a significant amelioration of atherosclerosis in APOE*3-Leiden.CETP mice. As shown in Figure 1A,B, representative cross sections of the aortic root of the WD control and WD + Oxy210-treated animals indicate significantly reduced lesion area with Oxy210 treatment, according to oil red O staining and lesion quantification by light microscopy. In addition, macrophage proliferation in the lesions was significantly reduced in the WD + Oxy210-treated animals compared to WD controls, according to imaging and quantification of Ki-67+ nuclei colocalized with CD68+ components and normalized to the total number of nuclei in CD68+-stained areas (Figure 1C,D). This was correlated with a decrease in total macrophage numbers determined by CD68 staining and quantification (Figure 1E,F). With this imaging methodology, a 50% decrease in proliferating macrophages (*p* < 0.05) was observed in Oxy210-treated mice as compared to the WD-fed control animals. This is consistent with the potent dose-dependent anti-proliferative effects of Oxy210 in RAW264.7 macrophages in vitro (Appendix A, IC_50_ = 0.88 ± 0.16 μM) and with its inhibitory effect on the expression of *M-CSF* in HAECs (see below), which is a major driver of macrophage proliferation during atherogenesis [57,58,59].

### 3.2. Oxy210 Lowers Plasma Cholesterol Levels, Liver Fibrosis, and Hepatic Profibrotic Gene Expression in APOE*3-Leiden.CETP Mice

High levels of circulating cholesterol, especially unesterified cholesterol, have been causally linked to atherogenesis. We previously reported that oral administration of Oxy210 to APOE*3-Leiden.CETP mice on a WD at 4 mg/g for 16 weeks produced significantly reduced plasma levels of total cholesterol and unesterified cholesterol at the end of the study [48]. However, circulating levels of TGs and free fatty acids were not significantly lowered by Oxy210 [48]. In the current study, we found that Oxy210 significantly reduced total cholesterol levels in a dose-dependent manner after 16 weeks of feeding at 0.5, 1, 2, and 4 mg/g formulated in WD (Figure 2A). However, unlike significant reductions in hepatic fibrosis (Figure 2B) and lowering of hepatic expression of fibrotic genes, including Acta2 and Spp1 (Figure 2C,D), that were caused at lower doses, Oxy210′s inhibitory effect on atherosclerotic lesion formation was only observed at the highest dose of 4 mg/g (Figure 2E). Moreover, in a separate study, we found that oral administration of Oxy210 to APOE*3-Leiden.CETP mice mixed in WD at 4 mg/g resulted in significant reductions in plasma total cholesterol, unesterified cholesterol, and LDL-C as early as 4 weeks after Oxy210 was provided to the mice (Figure 3). A transient trend in increased HDL levels was also found at 4 and 12 weeks, but not at 8 or 16 weeks of Oxy210 administration, which did not reach statistical significance (Appendix A).

### 3.3. Oxy210 Inhibits the Expression of Pro-Inflammatory and Atherogenic Genes by Oxidized Phospholipid, PGPC, in RAW264.7 Murine Macrophages

PGPC is a truncated oxidized phospholipid present in oxidatively modified low-density lipoprotein (Ox-LDL) that accumulates in atherosclerotic lesions [60,61,62]. PGPC was shown to impair vascular endothelial cell function and induce inflammatory responses in macrophages through inflammasome activation [63,64]. Treatment of RAW264.7 cells with PGPC for previously optimized time periods of 4 or 24 h for induction of maximum gene expression resulted in significantly increased expression of the pro-inflammatory and atherogenic genes, *JunB*, *Il-1β*, *c-Src,* and *Tnf-α*, and Oxy210 treatment inhibited the increased expression of these genes to near basal levels (Figure 4).

### 3.4. Oxy210 Inhibits TNF-α-Induced Expression of Vascular Cell Adhesion Molecule-1 (VCAM-1) in HAECs and Human THP-1 Macrophages

Increased production and release of TNF-α, a pro-inflammatory and pro-atherogenic cytokine, promotes atherosclerotic lesion formation and progression in part through the induced expression of adhesion molecules, including VCAM-1, on aortic endothelial cells, which results in increased binding, rolling, and recruitment of monocytes to the artery wall [65,66,67]. Treatment of primary HAECs with TNF-α resulted in a robust 185-fold induction of *VCAM-1* mRNA expression which was attenuated significantly in the presence of Oxy210 (Figure 5A). The pro-atherogenic effects of VCAM-1, however, are not limited to endothelial cells but have also been documented in macrophages through the mediation of metabolic changes and increased mitochondrial biogenesis and DNA oxidation, which result in increased inflammation and exacerbation of atherosclerosis [68,69]. Stimulation of human THP-1 macrophages with TNF-α resulted in an 80-fold induction of *VCAM-1* gene expression that was completely inhibited by Oxy210, returning the expression to basal levels (Figure 5B).

### 3.5. Oxy210 Inhibits PGPC- and CU-T12-9-Induced Expression of Pro-Inflammatory and Atherogenic Genes in HAECs

Under atherosclerotic conditions, the activation of vascular endothelial cells causes increased production of cytokines and chemokines, such as TNF-α and chemokine (C-C motif) ligand 2 (CCL2), as well as the over-expression of leukocyte adhesion molecules, such as VCAM-1 and selectins. Selectins mediate the first steps in the recruitment of monocytes from the blood stream in several pathologic situations, including atherosclerosis. The potential role of TLRs, including TLR2 and TLR4, in the pathogenesis of atherosclerosis and inflammatory effects of Ox-LDL has been reported [70,71,72,73]. To further examine the potential anti-inflammatory and anti-atherogenic effects of Oxy210 on vascular endothelial cells, we stimulated primary HAECs with the atherogenic oxidized phospholipid, PGPC, or the TLR2 agonist, CU-T12-9, in the presence or absence of Oxy210. Treatment of HAECs with PGPC resulted in increased mRNA expression of *M-CSF* that was significantly reduced by Oxy210 (Figure 6A). In addition, treatment of HAECs with CU-T12-9, a TLR2 agonist, increased the mRNA expression of inflammatory and atherosclerotic cytokines, *CCL2*, and *TNF-α*, as well as those of leukocyte adhesion molecules, *VCAM-1* and *E-Selectin*, all of which were significantly inhibited by Oxy210 (Figure 6B–E).

### 3.6. Oxy210 Inhibits TGF-β-Induced Expression of Pro-Inflammatory and Atherogenic Genes in HAECs

Paradoxically, depending on the cell type and context, TGF-β can display pro- or anti-inflammatory properties in addition to profibrotic properties [74]. TGF-β produced locally by vascular endothelial cells has been shown to exacerbate pro-inflammatory activities and chronic vascular inflammation in atherosclerotic lesions [75]. With systemic release, the immunosuppressive properties of TGF-β often predominate, mediated largely through anti-inflammatory effects in monocytes and macrophages. Figure 7 demonstrates primary HEACs treated with TGF-β in the presence or absence of Oxy210. The expression of pro-inflammatory genes *CCL2* and matrix metalloproteinase 2 (*MMP2*) was modestly but significantly induced in HAECs treated with TGF-β, and this induction, as well as their baseline levels, were inhibited by Oxy210 (Figure 7). By contrast, under these experimental conditions, the expression of *IL-6* or *TNF-α* was not significantly induced by TGF-β treatment, as shown in Appendix A. MMP2 is an enzyme that is involved in vascular extracellular matrix remodeling associated with reduced atherosclerotic plaque stability and mediates inflammatory responses in blood vessel wall [76,77].

### 3.7. Oxy210 Inhibits LPS-Induced Expression of Pro-Inflammatory and Atherogenic Genes in HAECs

Several disorders including obesity, NASH, insulin resistance, and type 2 diabetes can be associated with elevated LPS levels in circulation [78,79]. In primary HAECs, LPS-induced the expression of cell adhesion molecules *VCAM-1* and *E-Selectin* (Figure 8A,B), as well as pro-inflammatory genes, *IL-6*, *CCL2*, and *TNF-α* (Figure 8D–F). Oxy210 significantly inhibited these LPS-induced responses in a dose-dependent manner (Figure 8 and Appendix A). In good correlation with mRNA expression, LPS-induced VCAM-1 protein expression was also robustly inhibited by Oxy210 (Figure 8C). Even in the absence of LPS stimulation, protein expression of VCAM-1 was significantly reduced by Oxy210 treatment (Figure 8C). These findings are consistent with our previous report that Oxy210 inhibits LPS-induced inflammatory responses in macrophages that are mediated by TLR4 signaling [36].

### 3.8. Oxy210 Inhibits LPS-Induced Expression of Atherogenic Genes in HAECs under Prophylactic or Therapeutic Treatment Conditions

Next, we compared prophylactic treatment (i.e., before exposure to LPS) to therapeutic treatment (i.e., after exposure to LPS) of HAECs with Oxy210. As shown in Figure 9, the effects of Oxy210 on LPS-induced expression of atherogenic genes in HAECs were studied in four different experimental conditions: (A) 24 h of pretreatment with Oxy210, followed by addition of LPS for 4 h; (B) 24 h of pretreatment with Oxy210, followed by removal of Oxy210, and addition of LPS for 4 h; (C) 24 h of pretreatment with Oxy210, followed by removal of Oxy210 and continued cell culture for 24 h, followed by addition of LPS for 4 h; and (D) LPS treatment for 4 h followed by addition of Oxy210 for 24 h. Under all conditions, LPS treatment induced a robust expression of *VCAM-1* and *E-Selectin* compared to untreated control cells (see also Figure 8A), shown in Figure 9. Under pretreatment conditions (A–C), the inhibitory efficacy of Oxy210 on the expression of the genes remains nearly unchanged even if it was subsequently removed (A vs. B), unless the removal was prolonged for 24 h (A vs. C). Therapeutic treatment with Oxy210 (D) was nearly as effective as prophylactic treatment (A).

### 3.9. Oxy210 Enhances ABCA1 Abundance and Cholesterol Efflux in RAW264.7 Macrophages

The accumulation of cholesterol in macrophage foam cells is a central event in atherogenesis and results from the uptake of modified LDL-C deposited in the atherosclerotic lesion. Increased macrophage foam cell content is associated with disease progression in atherosclerosis and plaque instability. Macrophages are known to offload excess cholesterol to apolipoprotein A-I (ApoA-I) via the ATP-binding cassette transporter A1 (ABCA1), a cholesterol transport protein whose expression is regulated by liver X receptors (LXRs) [80]. Enhancement of such cholesterol efflux has been shown to be beneficial in the treatment of atherosclerosis and vulnerable plaque stabilization [81,82,83,84]. RAW264.7 macrophages treated with Oxy210 display significantly enhanced ABCA1 abundance (Figure 10A) associated with significantly increased cholesterol efflux to ApoA-I (Figure 10B). Of note, we had previously found Oxy210 to be devoid of significant LXR activity in HepG2 human hepatocyte and NIH3T3 fibroblast cells, as determined by the lack of induced expression of sterol regulatory element-binding protein 1c (SREBP1c) and ABCA1 genes [54], suggesting that the effects of Oxy210 on LXR signaling may be cell-type-specific.

## 4. Discussion

Dose response studies showed that the inhibitory effects of Oxy210 on atherosclerotic lesion formation were found only at the highest dose of 4 mg/g, whereas comparable improvements in liver fibrosis scores, profibrotic gene expression and lipid lowering were observed at lower doses of 0.5, 1, and 2 mg/g. Therefore, it is plausible that the anti-atherogenic effects of Oxy210 are not only due to its cholesterol-lowering properties but also caused by anti-inflammatory effects in artery wall cells. At the lower doses, Oxy210 concentrations achievable in the artery wall may not be sufficient to prevent atherosclerotic lesion formation, given the high liver-to-plasma ratio in Oxy210 concentrations encountered with oral dosing in WD ([48] and unpublished data). However, plasma exposure of Oxy210 in mice could be increased using traditional oral dosing (e.g., oral gavage) rather than dosing mixed in WD [54]. In addition to its anti-atherogenic effects in vivo, we demonstrate significant and broad-based anti-inflammatory and anti-atherogenic effects of Oxy210 in vitro, using cellular models of chronic inflammation and endothelial dysfunction that regulate atherogenesis, namely HAECs and macrophages. We demonstrate that Oxy210’s anti-atherogenic effects may be in part due to its ability to inhibit cellular responses that mediate the recruitment of monocytes into the vascular wall. Subsequently, this triggers the formation of fatty streaks that become advanced lesions in response to a sustained chronic inflammatory response by endothelial cells and macrophages. The negative regulation of adhesion molecule expression, including E-Selectin and VCAM-1, in endothelial cells, as well as inhibition of monocyte chemotactic factors, including CCL2, ultimately result in inhibition of monocyte trafficking. Oxy210 reduced macrophage proliferation in vitro and in vivo and dampened the expression of inflammatory markers in macrophages, induced by the exposure to proatherogenic agents, such as the atherogenic phospholipid, PGPC. Also noteworthy is the ability of Oxy210 to inhibit *M-CSF* expression. M-CSF is a secreted cytokine that can trigger hematopoietic stem cells to differentiate into macrophages and plays a significant role in macrophage proliferation and atherosclerotic plaque development [57,85]. Macrophages can also undergo clonal expansion within the atherosclerotic lesion [86]. During atherosclerotic plaque formation, tissue resident macrophages in the subendothelial space respond to inflammatory factors, including Ox-LDL and cytokines, and exacerbate inflammatory processes. The polyclonal expansion of macrophages has been identified as one of the mechanisms that drive atherogenesis and lesion formation [86]. In addition, macrophages treated with Oxy210 showed increased cholesterol efflux which may help to decrease macrophage foam cell content. In HAECs exposed to select atherogenic and pro-inflammatory agents, such as PGPC, TNF-α, CU, LPS, and TGF-β, Oxy210 reduced the expression of inflammatory markers (i.e., *IL-6*, *TNF-α*, *CCL2*, *VCAM-1*, and *E-Selectin*). In cultured HAECs, we demonstrate that Oxy210 can inhibit the high baseline and TGF-β-induced expression of *CCL-2* and *MMP2*, further supporting its potential utility in the context of vascular inflammation and atherosclerosis. We have previously shown that the effects of Oxy210 on TGF-β signaling are cell-type-specific: Oxy210 does not interfere with the anti-inflammatory effects of TGF-β and TGF-β-induced Smad2/3 phosphorylation in macrophages [36], whereas it does inhibit TGF-β-induced pro-fibrotic effects in hepatic stellate cells, kidney mesangial cells, and lung fibroblasts [48,87].

Collectively, these findings support the hypothesis that a combination of cholesterol-lowering and anti-inflammatory effects of Oxy210 may mediate its anti-atherogenic effects. This mechanism would be distinct from existing therapies that exert anti-atherogenic effects through either direct cholesterol lowering (e.g., statins) or direct anti-inflammatory effects (e.g., Canakinumab).

As noted earlier, several disorders, including obesity, NASH, insulin resistance, and type 2 diabetes, are often associated with elevated LPS levels in circulation, linked to pro-inflammatory TLR signaling [78,79]. Obesity is known to increase intestinal permeability (leaky gut syndrome), which allows LPS and other factors derived from gut bacteria to spill over into the portal blood circulation and liver [88]. Impaired liver-clearance functions, often a result of NAFLD/NASH, then permit LPS to enter the systemic circulation. Even at low levels, LPS can have a harmful impact on the endothelium [89] and drive low-grade chronic inflammation in visceral organs and fat tissue [90]. Oxy210 treatment may dampen such deleterious effects mediated by LPS according to data presented here and in previous studies [36,48]. In experimental settings using LPS, the remedial effects of anti-inflammatory drugs can often be enhanced by prophylactic treatment (i.e., drug administration before LPS treatment), whereas therapeutic treatment (i.e., drug administration after LPS treatment) can be less effective [91,92]. Moreover, the efficacy of drug treatments may be transient and/or require the presence of the drug throughout the LPS treatment period, and its removal may diminish or eliminate the therapeutic benefit. It is noteworthy that the inhibitory effect of Oxy210 in HAECs on the LPS-induced expression of atherogenic and inflammatory genes occurred almost to the same degree under prophylactic and therapeutic treatment conditions. Furthermore, these inhibitory effects prevailed even after removal of Oxy210, although we cannot completely rule out the possibility that Oxy210 may have been sequestered inside the cells after removal from the tissue culture media. Since the therapeutic treatment was nearly as effective as prophylactic treatment in these in vitro experiments, in future studies, we will assess the therapeutic effects of Oxy210 in mouse models of atherosclerosis and NASH. Such experimental settings could be more relevant to the human condition, as most, if not all, humans suffering from atherosclerosis and NASH have a certain degree of established disease upon diagnosis.

Unlike traditional drug candidates, designed to target a specific enzyme or pathway, Oxy210 displays inhibitory properties across multiple signaling pathways in a cell-type-specific manner, observed in cell culture of fibroblasts (Hh and TGF-β signaling [48,54,87]), macrophages (TLR signaling [36]), epithelial cells (Hh, TGF-β, and Wnt signaling, [54,87,93]), and endothelial cells (inflammatory and TGF-β signaling). Even though these pathways are notable for their involvement in chronic inflammatory and fibrotic diseases, it may seem implausible for Oxy210 to produce multiple pathway inhibition by engaging a single target protein, (unless situated up-stream from the affected signaling pathways). While promiscuous protein binding of drug molecules can result in polypharmacology, leading to off-target effects [94], we have previously shown that Oxy210 is conformationally rigid and that close structural analogs of Oxy210, such as Oxy43 and Oxy186, display strikingly different properties [36,54] which would argue against indiscriminate protein affinity. On the other hand, it has been reported that Hh [95], TGF-β [96], Wnt [97], and TLR signaling [98] are all constrained and regulated within the context of (well-functioning) cholesterol homeostasis and that the pathological activation of these signaling pathways requires availability and release of excess cellular cholesterol. As some endogenous oxysterols are known regulators of cholesterol homeostasis [43,99,100], this may point to a possible mechanistic origin for the diverse biological properties of Oxy210 along with potential therapeutic implications in NASH and atherosclerosis, all of which are currently being evaluated.

Given the multifactorial pathogenesis and partial overlap between NASH and atherosclerosis, combination therapies with agents that individually target different disease-driving pathways have been explored. However, combination therapies can add to the complexities and cost of clinical trials, and, at least in the context of NASH, have not yet proven more effective or safer compared to monotherapy approaches [101,102]. Alternatively, a “designed polypharmacology approach”, aiming to fuse complementary properties into a single molecule, could be a productive therapeutic strategy. According to a recently published report, using a triple agonist, designed to combine FXR activation with dual peroxisome-proliferator-activated receptor (PPAR α and δ) activation, in diet-induced mouse models of NASH, produced superior outcomes compared to mono-targeted agents [103]. By combining antifibrotic and anti-inflammatory activities with lipid-lowering properties, Oxy210 may fall into a conceptually similar category.

## 5. Conclusions

Altogether, our findings to date suggest that oxysterol analogs like Oxy210 could be considered as new drug candidates for targeting both atherosclerosis and NASH, as well as chronic inflammatory conditions associated with metabolic syndrome and inflammaging.

## Figures and Tables

**Figure 1 cells-13-01632-f001:**
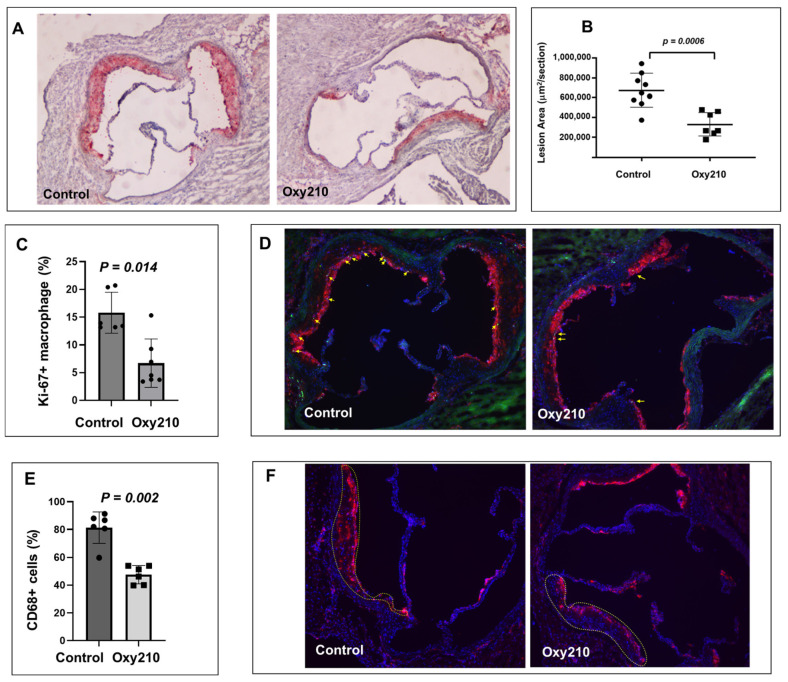
Inhibition of atherosclerotic lesion formation and macrophage proliferation by Oxy210 in atherosclerotic lesions in APOE*3-Leiden.CETP mice. Control WD (*n* = 9) and WD + Oxy210-treated (*n* = 7) mice were euthanized and the upper portion of the heart and proximal aorta were embedded in OCT compound and stored at −70 °C. Serial 10 μm-thick cryosections from the middle portion of the left ventricle of the aortic root were collected and mounted on poly-d-lysine-coated plates. Sections were stained with oil red O. Representative pictures (**A**) are shown at magnification of 4X. Lesion areas (**B**) were quantitated by light microscopy and ImagePro Premier software (Version 9.1). Results are presented as mean ± SEM. To quantitate macrophage proliferation (**C**,**D**), immunofluorescence staining was performed using a standard protocol. Merged-CD68 (red), Ki-67 (green), and DAPI (blue), shown at magnification of 10X. Yellow arrows represent Ki-67+ macrophages. The percentage of CD68+ macrophages (**E**,**F**) relative to total cells (DAPI+ cells) in atherosclerotic lesions (outlined) was measured in both control WD mice and those treated with WD + Oxy210. WD control mice exhibit a significantly higher percentage of macrophages in the atherosclerotic lesions compared to Oxy210-treated mice, where macrophage infiltration is notably reduced. Merged-CD68 (red) and DAPI (blue), shown at magnification of 10X.

**Figure 2 cells-13-01632-f002:**
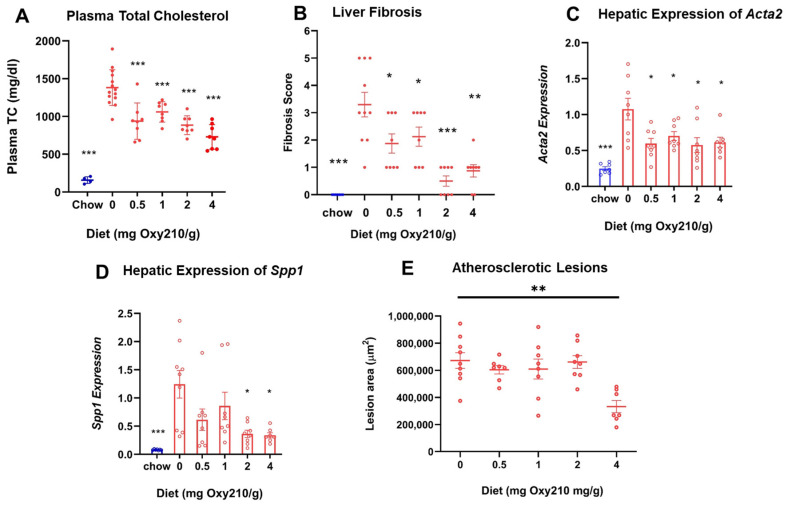
Dose-dependent effects of Oxy210 on plasma total cholesterol, liver fibrosis, hepatic profibrotic gene expression, and atherosclerotic lesion formation. (**A**) Mice were fed a chow diet or WD supplemented with various amounts of Oxy210 for 16 weeks. Mice were fasted for 4 h prior to blood collection. Plasma total cholesterol (TC) was measured by colorimetric assays. Data are presented as mean ± SD (*n* = 7–14 mice per group). *** denotes *p* < 0.001 versus control (WD without Oxy210 supplementation). (**B**) Liver sections from control and Oxy210-fed mice (*n* = 7–10 animals per group) were stained for collagen with picrosirius red. Fibrosis score was determined by a pathologist blinded to the study. Results are presented as mean ± SEM. * denotes *p* < 0.05, ** denotes *p* < 0.01, and *** denotes *p* < 0.01 versus control (WD without Oxy210 supplementation). The expression of pro-fibrotic genes *Acta2* (**C**) and *Spp1* (**D**) in the livers from chow control, WD-, and WD Oxy210-fed mice (0.5, 1, 2, and 4 mg/g) was measured by qPCR and normalized to the level of the housekeeping gene *Rpl4*. Relative gene expression levels are presented as mean ± SD in each group (*n* = 6–8). * denotes *p* < 0.05 versus control. (**E**) The measurement of atherosclerotic lesions was conducted through the examination of sections obtained from the aortic sinus and proximal aorta of mice subjected to varying doses of Oxy210.

**Figure 3 cells-13-01632-f003:**
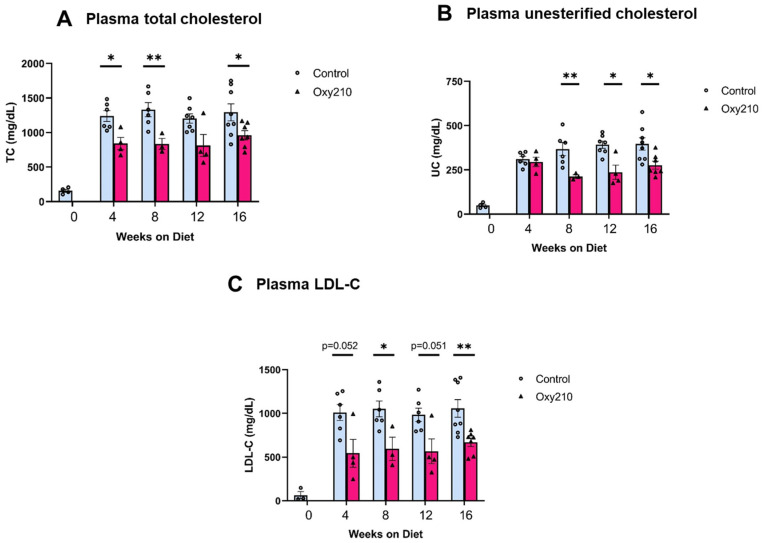
Time course of lipid-lowering effects of Oxy210. Female CETP/APOE*3-Leiden mice were fed WD with (purple bars) or without (blue bars) supplementation of Oxy210 (4 mg/g) for 0–16 weeks. Plasma samples were collected after a 4 h fast at the indicated time points. Levels of (**A**) total cholesterol (TC), (**B**) unesterified cholesterol (UC), and (**C**) LDL cholesterol (LDL-C) were determined using colorimetric assays. Results from control WD and WD + Oxy210 mice are presented as mean ± SEM from each group (*n* = 3–7). * denotes *p* < 0.05 and ** denotes *p* < 0.01.

**Figure 4 cells-13-01632-f004:**
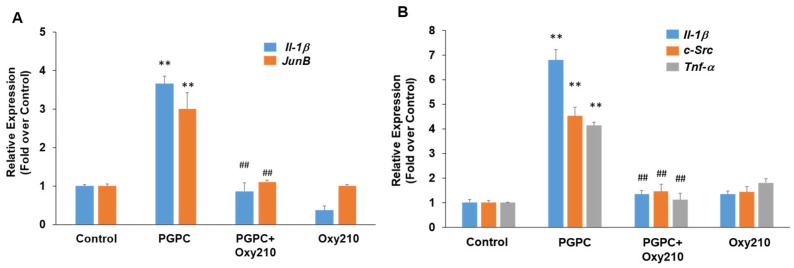
Inhibition of PGPC-induced expression of pro-inflammatory and atherogenic genes in RAW264.7 cells by Oxy210. RAW264.7 cells were treated in DMEM containing 0.1% FBS overnight and then pretreated for 2 h with Oxy210 (5 μM) in DMEM containing 0.1% FBS. The cells were then treated with PGPC (50 μM) in the absence or presence of Oxy210 (5 μM). After 4 (**A**) or 24 (**B**) hours, RNA was extracted and analyzed by Q-RT-PCR for the expression of the genes as indicated and normalized to Oaz1 expression. Data from a representative experiment are reported as the mean of triplicate determinations ± SD (## *p* < 0.01 vs. PCPG; ** *p* < 0.01 vs. control).

**Figure 5 cells-13-01632-f005:**
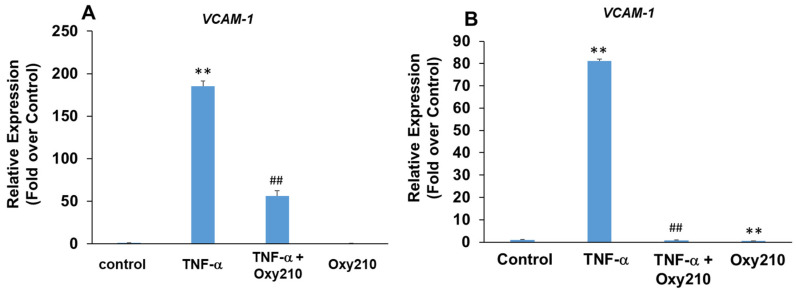
Inhibition of TNF-α-induced expression of *VCAM-1* in HAECs and THP-1 cells by Oxy210. (**A**) The HAECs were pretreated with Oxy210 (5 μM) in M199 containing 1% FBS overnight and then treated with TNF-α (50 ng/mL) in the absence or presence of Oxy210 (5 μM). After 8 h, the RNA was extracted and analyzed by Q-RT-PCR for the expression of VCAM-1 and normalized to GAPDH expression. Data from a representative experiment are reported as the mean of triplicate determinations ± SD (## *p* < 0.01 vs. TNF-α; ** *p* < 0.01 vs. control). (**B**) The THP-1 cells were pretreated with Oxy210 (5 μM) in RPMI containing 1% FBS overnight and then treated with TNF-α (50 ng/mL) in the absence or presence of Oxy210 (5 μM). After 24 h, the RNA was extracted and analyzed by Q-RT-PCR for the expression of VCAM-1 and normalized to GAPDH expression. Data from a representative experiment are reported as the mean of triplicate determinations ± SD (## *p* < 0.01 vs. TNF-α; ** *p* < 0.01 vs. control).

**Figure 6 cells-13-01632-f006:**
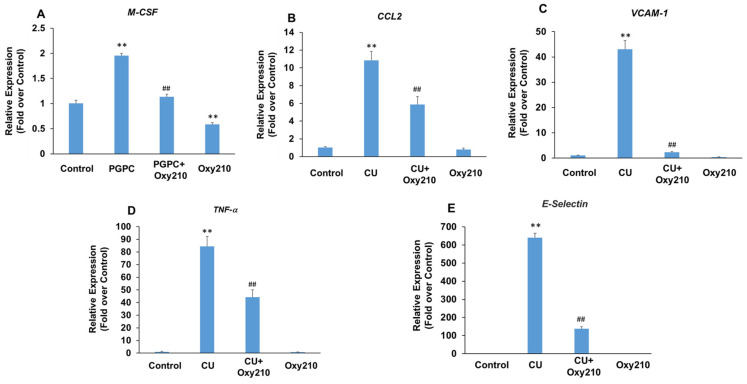
Inhibition of PGPC and CU-T12-9-induced expression of pro-inflammatory and atherogenic genes in HAECs by Oxy210. HAECs were pretreated with (**A**) Oxy210 (5 μM) or (**B**–**E**) Oxy210 (10 μM) in M199 containing 1% FBS overnight and then treated with 50 μM of (**A**) PGPC or (**B**–**E**) 1 μg/mL of CU-T12-9 (CU) in the absence or presence of Oxy210. After (**A**) 24 or (**B**–**E**) 4 h, the RNA was extracted and analyzed by Q-RT-PCR for the expression of the genes as indicated and normalized to GAPDH expression. Data from a representative experiment are reported as the mean of triplicate determinations ± SD (## *p* < 0.01 vs. PCPG or CU; ** *p* < 0.01 vs. control).

**Figure 7 cells-13-01632-f007:**
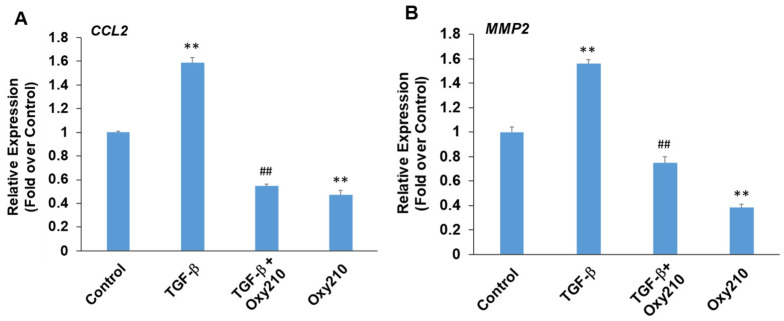
Inhibition of TGF-β-induced expression of pro-inflammatory and atherogenic genes in HAECs by Oxy210. HAECs were pretreated with Oxy210 (5 μM) in M199 containing 1% FBS for 6 h and then treated with rhTGF-β1 (10 ng/mL) in the absence or presence of Oxy210. After 48 h, RNA was extracted and analyzed by Q-RT-PCR for the expression of the genes as indicated and normalized to GAPDH expression. Data from a representative experiment are reported as the mean of triplicate determinations ± SD (## *p* < 0.01 vs. TGF-β; ** *p* < 0.01 vs. control).

**Figure 8 cells-13-01632-f008:**
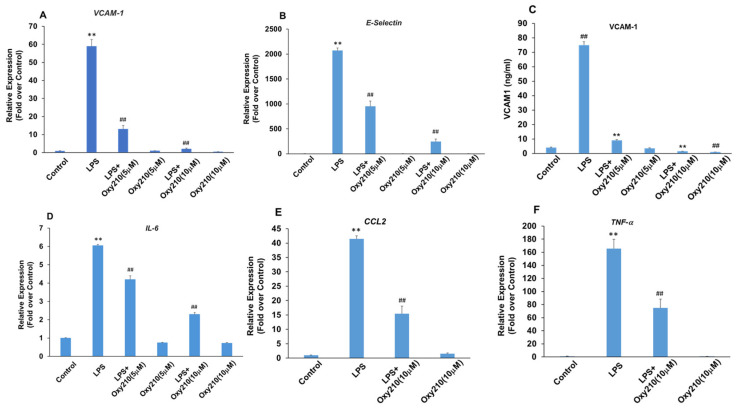
Inhibition of LPS-induced expression of atherogenic and pro-inflammatory factors in HAECs by Oxy210. (**A**,**B**,**D**–**F**) HAECs were pretreated with Oxy210 as indicated in M199 containing 1% FBS overnight and then treated with 1 μg/mL of LPS in the absence or presence of Oxy210. After 4 h, RNA was extracted and analyzed by Q-RT-PCR for the expression of the genes as indicated and normalized to GAPDH expression. Data from a representative experiment are reported as the mean of triplicate determinations ± SD (## *p* < 0.01 vs. LPS; ** *p* < 0.01 vs. control). (**C**) HAECs were pretreated with increasing concentrations of Oxy210 in M199 containing 1% FBS for 24 h and then treated with LPS (1 μg/mL) in the absence or presence of Oxy210 as indicated. After 24 h, cells were lysed, and the whole-cell extracts were diluted and subjected to ELISA for VCAM-1. Data from a representative experiment are reported as the mean of triplicate determinations ± SD (## *p* < 0.01 vs. control; ** *p* < 0.01 vs. LPS).

**Figure 9 cells-13-01632-f009:**
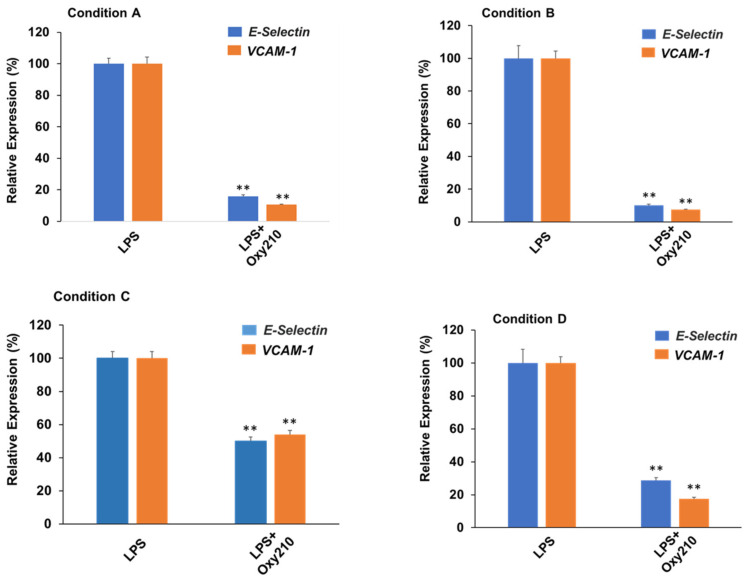
Prophylactic and therapeutic inhibition of LPS-induced expression of atherogenic genes in HAECs by Oxy210. HAECs were treated under the following four experimental conditions where cells were exposed to Oxy210 (5 μM) before (prophylactic) or after (therapeutic) treatment with LPS (1 μg/mL): (**A**) 24 h of pretreatment with Oxy210, followed by addition of LPS for 4 h; (**B**) 24 h of pretreatment with Oxy210, followed by removal of Oxy210, and addition of LPS for 4 h; (**C**) 24 h of pretreatment with Oxy210, followed by removal of Oxy210 and continued cell culture for 24 h, followed by addition of LPS for 4 h; and (**D**) LPS treatment for 4 h followed by addition of Oxy210 for 24 h. RNA was extracted and analyzed by Q-RT-PCR for the expression of the genes as indicated and normalized to GAPDH expression. Data from a representative experiment are reported as the mean of triplicate determinations ± SD (** *p* < 0.01 vs. LPS).

**Figure 10 cells-13-01632-f010:**
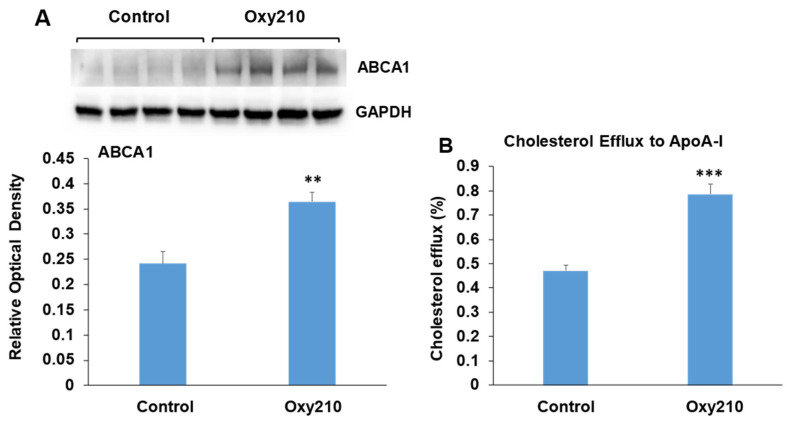
Stimulation of ABCA1 protein abundance and cholesterol efflux by Oxy210 in RAW264.7 macrophages. (**A**) Raw264.7 cells were grown to confluence in the DMEM containing 10% FCS and then incubated with Oxy210 (5 μM) or vehicle for 24 h in medium containing 0.1% FCS. Cells were then lysed, and ABCA1 and GAPDH in lysates were detected by Western blotting and visualized and quantitated using G:Box imaging system. Data from a representative experiment are reported (*n* = 6, biological replicates) as mean ± SD (** *p* < 0.01 vs. control). (**B**) Raw264.7 cells were grown to confluence and labeled with [3H] cholesterol for 48 h in a medium containing 10% FCS. Cells were then incubated for 24 h with Oxy210 (5 μM) or vehicle in a medium containing 0.1% FCS. Cells were then washed and incubated for 2 h in a serum-free medium containing purified human apoA-I (25 μg/mL) or without apoA-I as control. Specific cholesterol efflux is shown and was expressed as a percentage of labeled cholesterol moved from cells to apoA-I-containing media minus that in the absence of apoA-I (blank). Data from a representative experiment are reported (*n* = 6, biological replicates) as mean ± SEM (*** *p* < 0.001 vs. control).

## Data Availability

The data presented in this study are available on request and within reason from the corresponding author (F.P.). The data are not publicly available due to privacy.

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
