# Peer review of "Anti-Inflammatory Oxysterol, Oxy210, Inhibits Atherosclerosis in Hyperlipidemic Mice and Inflammatory Responses of Vascular Cells"

_cells, 2024, doi:10.3390/cells13191632_

Round 1
Reviewer 1 Report
Comments and Suggestions for Authors
The article “Anti-inflammatory oxysterol, Oxy210, inhibits atherosclerosis in hyperlipidemic mice and inflammatory responses of vascular cells” demonstrated that a previous reported oxysterol Oxy210 in a humanized mouse model for both NASH and atherosclerosis is able to reduce atherosclerotic lesion formation in a mice feed with western diet. In vitro cell line experiments, Oxy210 was able to reduced the expression of some inflammatory markers
Major revision
Line 62: please define what should be “ideal”
Material and methods:
2.8. Quantification of Atherosclerotic Lesions and Macrophage Proliferation.(line 188)
CD68 is a scavenger receptor expressed highly by tissue macrophages and generally considered a pan-macrophage marker. It is a intracellular glycoprotein primarily reported to be associated with cytoplasmic granules and to a lesser extent the membranes of macrophages. I could not find nuclear description of CD68. Please make sure the description is correct and the result either.
Results
The results presented are always contextualized with previous results and literature, but it is confusing to understand what was done previously with what is being presented and some texts could be part of discussion and not results.
Ex.: line 254 - It is difficult to understand if the result is that of this article. Please make it more clear in all text
Line 266-274: You used different times and in those times different genes were made. What is the rationale for this? the different times are not explicit in the text, only in the graphic
Line 326: IL-6 and TNF- α was not present in figure. If mentioned please show the data
Line 355 – 360: It is a theoretical explanation and should be removed
Line 361: figure 4 is the base of this results it should not be supplementary but a result
Discussion
The discussion presents data already presented in the introduction and results without presenting much new information. The results are presented but not discussed. Much of the discussion was done directly in the results section and this should be rearranged. Discussion must be rewritten.
Figures should be positioned close to the text to which they refer
Figures legends
Figure 1 - Sometimes it is used CD68 and other times macrophage. Please use always the same description.
Please present the imunofluorecence images with arrows to the co-localization of CD68 and Ki67.
Do not repeat methodology in the legend figure.
Author Response
Our team of authors is grateful to the reviewers for their valuable comments and suggestions. We are pleased to submit our revised manuscript in which we have addressed the reviewers’ suggestions, and the following are our point-by-point responses to their comments.
Reviewer 1
Introduction
- Line 62: please define what should be “ideal”
Response: By “ideal”, we meant “therapeutically most effective”. We have removed the repeated use of the word in the revised manuscript and clarified our intended message.
Material and methods
- 8. Quantification of Atherosclerotic Lesions and Macrophage Proliferation. (line 188) CD68 is a scavenger receptor expressed highly by tissue macrophages and generally considered a pan-macrophage marker. It is an intracellular glycoprotein primarily reported to be associated with cytoplasmic granules and to a lesser extent the membranes of macrophages. I could not find nuclear description of CD68. Please make sure the description is correct and the result either.
Response: We appreciate this comment that caused the confusion in the way the statement was written. We have corrected the corresponding passage in the revised manuscript to:
“The quantification of Ki67+ staining was done by normalizing the Ki67+ nuclei colocalized with intracellular and membrane associated CD68+ components to total nuclei numbers in CD68+ stained area in lesions using a Nikon (Eclipse Ti-s) microscope.”
- Theresults presented are always contextualized with previous results and literature, but it is confusing to understand what was done previously with what is being presented and some texts could be part of discussion and not results. Ex.: line 254 - It is difficult to understand if the result is that of this article. Please make it more clear in all text
Response: We thank the reviewer for this important feedback and have adjusted the revised manuscript accordingly. Whenever possible, we refer to current data with comments like “Here”, “In this study”, or “In the current study”. Older data is referred to by comments like " We previously reported” and marked by references, “[ref]”.
- Line 266-274: You used different times and in those times different genes were made. What is the rationale for this? the different times are not explicit in the text, only in the graphic.
Response: In the revised manuscript we have clarified that the different times were used based on previously optimized data for the maximum induction of the genes by PGPC in RAW264.7 cells.
- Line 326: IL-6 and TNF- α was not present in figure. If mentioned please show the data
Response: As suggested by the reviewer, we have now provided the IL-6 and TNF- α data in supplemental Figure 4 in the revised manuscript. The expression of IL-6 and TNF- α was not significantly induced by TGF-b in HAECs.
- Line 355 – 360: It is a theoretical explanation and should be removed.
Response: We respectfully disagree with the reviewer on our explanation being “theoretical” in the context of drug screening, pharmacology and therapeutic efficacy. We have clarified this explanation in the revised manuscript in the Discussion section and have provided relevant literature.
- Line 361: figure 4 is the base of this results it should not be supplementary but a result
Response: As suggested by the reviewer, we have moved the content from the Supplemental Figures to the main Figures, now Figure 9 in the revised manuscript.
Discussion
- The discussion presents data already presented in the introduction and results without presenting much new information. The results are presented but not discussed. Much of the discussion was done directly in the results section and this should be rearranged. Discussion must be rewritten.
Response: As requested by the reviewer, in the revised manuscript, we have reorganized and rewritten large parts of the Introduction and Discussion Sections and, as suggested, moved content from the Results Section into the Discussion Section.
- Figures should be positioned close to the text to which they refer
Response: We have used the Cells template in the preparation of the manuscript. The Figures were placed in the Results Section according to the instructions of the Cells template.
Figures legends
- Figure 1 - Sometimes it is used CD68 and other times macrophage. Please use always the same description.
Response: We appreciate this comment by the reviewer and have made appropriate changes in the revised manuscript.
- Please present the immunofluorescence images with arrows to the co-localization of CD68 and Ki67.
Response: We thank the reviewer for this suggestion and have added immunofluorescence images to Figure 1.
- Do not repeat methodology in the legend figure.
Response: We have made appropriate changes in the revised manuscript.

Reviewer 2 Report
Comments and Suggestions for Authors
Oxy210, an oxysterol derivative, was originally reported to inhibit TGF-b and Hedgehog signaling, thereby inhibiting cancer cell proliferation. The drug candidate also ameliorated NASH in APOE*3.Leiden mice. In this manuscript, the authors showed that Oxy210 attenuated the development of atherosclerosis in APOE*3.Leiden.CETP mice fed a Western diet. The compound also demonstrated a variety of pharmacological effects favorable to anti-atherogenicity in human aortic endothelial cells and other cell lines. Concerns are listed below:
Concerns:
1) Oyx210 affected all cell lines studied to date, which may influence the potential systemic effect in mice. The authors should consider providing data to negate potential adverse effects in mice, such as body weight and toxicity after long-term treatment or when exposed to high doses of Oxy210. The authors should also consider demonstrating cell viability using assays such as ATP and XTT.
2) The first two paragraphs of the Discussion are more of a drug development concept. These can be shortened or the shorter version can be moved to the Introduction or the following paragraph of the Discussion.
Alternatively, they should discuss the pharmacological effects of Oxy210. Together with the authors' previous work, Oxy210 appears to be a versatile compound. In this manuscript, the results of all of the experiments performed in this work are in favor of anti-atherosclerosis, although the cells used and the readouts differ each other. Can these effects, or parts of them, be explained by the modification of TGF-beta signaling? For example, the primary cellular effect of Oxy210 appears to be the attenuation of TGF-beta signaling. The change in TLR signaling in the macrophage line may be due to crosstalk between the two signals (e.g., https://www.ncbi.nlm.nih.gov/pmc/articles/PMC8547806/). The readers may expect that the authors discuss the role of TGF-beta signaling in atherosclerosis and the effect of Oxy210 in this context.
3) How can Oxy210 lower cholesterol? Does oxy210 act as an LXR ligand?
4) In Figure 1, should the number of Ki67-positive macrophages be presented as Ki67 per CD68?
5) The supplementary figures and a table are included in the manuscript. They should be removed.
6) L254., Should "In a more recent study" be "In the current study"?
7) The authors should consider the placement of the figures to be readable.
Author Response
Our team of authors is grateful to the reviewers for their valuable comments and suggestions. We are pleased to submit our revised manuscript in which we have addressed the reviewers’ suggestions, and the following are our point-by-point responses to their comments.
Reviewer 2:
Comments and Suggestions for Authors
Concerns:
- Oyx210 affected all cell lines studied to date, which may influence the potential systemic effect in mice. The authors should consider providing data to negate potential adverse effects in mice, such as body weight and toxicity after long-term treatment or when exposed to high doses of Oxy210. The authors should also consider demonstrating cell viability using assays such as ATP and XTT.
Response: We thank the reviewer for raising these important concerns. In our cell culture work, including all data presented in this manuscript, we routinely monitor for morphological changes and cell viability via microscopy. As we develop Oxy210 as a drug candidate, we are keenly interested in the drug safety profile of Oxy210. Accordingly, we have performed numerous studies with Oxy210 in the areas of drug safety, toxicology and in vitro pharmacology (e.g., maximum tolerated dose and reproductive toxicity testing in mice; in vitro hERG testing, receptor screening etc). These studies are beyond the scope of the current manuscript which is more narrowly focused on demonstrating the efficacy of Oxy210 in inhibiting atherosclerosis in hyperlipidemic mice and inflammatory responses of vascular cells. We have added additional safety information in section 2.6 of the revised manuscript. Additional IND-enabling safety studies are planned and will be conducted according to FDA guidelines for new chemical entities.
- The first two paragraphs of the Discussion are more of a drug development concept. These can be shortened or the shorter version can be moved to the Introduction or the following paragraph of the Discussion. Alternatively, they should discuss the pharmacological effects of Oxy210. Together with the authors' previous work, Oxy210 appears to be a versatile compound. In this manuscript, the results of all of the experiments performed in this work are in favor of anti-atherosclerosis, although the cells used and the readouts differ each other. Can these effects, or parts of them, be explained by the modification of TGF-beta signaling? For example, the primary cellular effect of Oxy210 appears to be the attenuation of TGF-beta signaling. The change in TLR signaling in the macrophage line may be due to crosstalk between the two signals (e.g., https://www.ncbi.nlm.nih.gov/pmc/articles/PMC8547806/). The readers may expect that the authors discuss the role of TGF-beta signaling in atherosclerosis and the effect of Oxy210 in this context.
Response: We appreciate these excellent suggestions by the reviewer and have considered them in the revised manuscript. As suggested, we have moved drug development concepts to the Introduction and rewritten the Discussion with particular emphasis on how Oxy210 may be helpful in atherosclerosis with respect to LPS and TGF-beta signaling.
- How can Oxy210 lower cholesterol? Does oxy210 act as an LXR ligand?
Response: This question lies at the very core of our ongoing research. While oxysterols are known as LXR ligands, they are partial agonists that seldom fully activate the pathway to the same extent as synthetic ligands that are full agonists, such as T0901317 and GW3965. We have examined Oxy210 with respect to the activation/inhibition of LXR target genes (e.g., ABCA1, SREBP-1c) in various cell types. In some cell types, such as THP-1 macrophages and A549 cells, the expression of LXR target genes (ABCA1) was mildly induced, whereas in other cell types, such as HepG2 cells, some LXR target genes (ABCA1 and SREBP-1c) were inhibited whereas ABCG1 was induced. Although still elusive, we believe the cholesterol lowering effect of Oxy210 is through a mechanism distinct from that of statins.
- In Figure 1, should the number of Ki67-positive macrophages be presented as Ki67 per CD68?
Response: We have revised the manuscript to clarify this point. Since the purpose of the assessment was to determine the effect of Oxy210 treatment on macrophage proliferation, the number of Ki67+CD68+ cells (i.e. proportion of proliferating macrophages) normalized to total number of CD68+ cells.
- The supplementary figures and a table are included in the manuscript. They should be removed.
Response: We will submit the supplementary materials as a separate document for the submission of the finalized manuscript. For review purposes, we decided to keep the main and supplementary Figures in the same document to make it easier for the reviewers to refer to them while reviewing the manuscript.
- , Should "In a more recent study" be "In the current study"?
Response: Yes. This is important feedback, and we have adjusted the revised manuscript accordingly. Whenever possible, we refer to current data with comments like “Here”, “In this study”, or “In the current study”. Older data is referred to by comments like " We previously reported” and marked by references, “[ref]”.
- The authors should consider the placement of the figures to be readable.
Response: We have used the Cells template in the preparation of the manuscript. The Figures were placed in the Results Section according to the instructions of the Cells template.

Round 2
Reviewer 1 Report
Comments and Suggestions for Authors
No more concerns about the manuscript. It is suitable for publication in the form presented
Reviewer 2 Report
Comments and Suggestions for Authors
The authors have fully answered my concerns. The authors will not show the toxicity related data in this manuscript. I hope that the data will be published soon.